# Macrophage scavenger receptor 1 controls Chikungunya virus infection through autophagy in mice

Long Yang[1,12], Tingting Geng[2,12], Guang Yang[1,3,12], Jinzhu Ma[1], Leilei Wang[1], Harshada Ketkar[1], Duomeng Yang[2], Tao Lin[2], Jesse Hwang[4], Shu Zhu [5,11], Yanlin Wang[6], Jianfeng Dai[7], Fuping You[8], Gong Cheng [9], Anthony T. Vella[2], Richard. A. Flavell [5,10✉], Erol Fikrig [4,10✉] & Penghua Wang [1,2✉]

Macrophage scavenger receptor 1 (MSR1) mediates the endocytosis of modified low-density lipoproteins and plays an important antiviral role. However, the molecular mechanism underlying MSR1 antiviral actions remains elusive. We report that MSR1 activates autophagy to restrict infection of Chikungunya virus (CHIKV), an arthritogenic alphavirus that causes acute and chronic crippling arthralgia. Msr1 expression was rapidly upregulated after CHIKV infection in mice. *Msr1* knockout mice had elevated viral loads and increased susceptibility to CHIKV arthritis along with a normal type I IFN response. Induction of LC3 lipidation by CHIKV, a marker of autophagy, was reduced in $Msr1^{-/-}$ cells. Mechanistically, MSR1 interacted with ATG12 through its cytoplasmic tail and this interaction was enhanced by CHIKV nsP1 protein. MSR1 repressed CHIKV replication through ATG5-ATG12-ATG16L1 and this was dependent on the FIP200-and-WIPI2-binding domain, but not the WD40 domain of ATG16L1. Our results elucidate an antiviral role for MSR1 involving the autophagic function of ATG5-ATG12-ATG16L1.

[1] Department of Microbiology & Immunology, School of Medicine, New York Medical College, Valhalla, NY 10595, USA. [2] Department of Immunology, School of Medicine, University of Connecticut Health Center, Farmington, CT 06030, USA. [3] Department of Parasitology, School of Medicine, Jinan University, Guangzhou, 510632, China. [4] Section of Infectious Diseases, Yale University School of Medicine, New Haven, CT 06520, USA. [5] Department of Immunobiology, Yale University School of Medicine, New Haven, CT 06520, USA. [6] Department of Medicine, School of Medicine, University of Connecticut Health Center, Farmington, CT 06030, USA. [7] Institutes of Biology and Medical Sciences, Soochow University, Jiangsu, China. [8] School of Basic Medical Sciences, Peking University, Beijing, China. [9] Department of Basic Sciences, School of Medicine, Tsinghua University, Beijing, China. [10] Howard Hughes Medical Institute, Chevy Chase, MD, USA. [11]Present address: Hefei National Laboratory for Physical Sciences at Microscale, the CAS Key Laboratory of Innate Immunity and Chronic Disease, School of Basic Medical Sciences, Division of Life Sciences and Medicine, University of Science and Technology of China, Hefei 230027, China. [12]These authors contributed equally: Long Yang, Tingting Geng, Guang Yang. ✉email: Richard.flavell@yale.edu; erol.fikrig@yale.edu; Pewang@uchc.edu

Macrophage scavenger receptor 1 (MSR1) is a member of the scavenger receptor family, members of which are structurally heterogeneous with little or no homology. They are grouped into the same family due to their shared functional properties such as the ability to bind and internalize a diverse range of self- and non-self-ligands including modified low-density lipoproteins (LDL), lipopolysaccharide (LPS), and lipoteichoic acid (LTA)[1]. The ligand promiscuity of these scavenger receptors is probably due to their function as components of other pathogen pattern recognition receptor (PRR) signaling complexes. For example, MSR1 interacts with toll-like receptor (TLR) 4 and TLR2 to promote phagocytosis of Gram-positive and negative bacteria, respectively[2], and is required for LPS-induced TLR4 signaling[3]. However, the current literature about the physiological function of MSR1 in viral infections is inconsistent. MSR1 could recognize extracellular viral nucleic acids, mediating their endocytosis and presenting them to TLR3/9, thereby triggering a type I interferon (IFN-I) response in both infected and uninfected cells[4,5]. MSR1 limits herpes simplex virus 1 infection likely by regulating macrophage adherence in vivo[6], whereas it controls adenovirus type 5-elicited hepatic inflammation and fibrosis by promoting M2 macrophage polarization[7]. MSR1 could contribute to the pathogenesis of murine hepatitis virus-induced fulminant hepatitis by activating neutrophil-mediated complement pathways[8]. Ectopic expression of MSR1 inhibited Sindbis virus A and human parainfluenza virus type 3 in $STAT1^{-/-}$ fibroblasts[9], suggesting an IFN-I-independent antiviral role for MSR1. These contrasting results suggest that the antiviral mechanisms of MSR1 vary with viral species and disease models. However, the molecular mechanism underlying MSR1 antiviral actions remains elusive.

Alphaviruses belong to a family of positive sense, single-stranded RNA viruses that cause arthritis and/or encephalitis in humans and animals. For example, chikungunya virus (CHIKV) causes acute and chronic crippling arthralgia and long-term neurological disorders. Since 2005, following several decades of relative silence, CHIKV has re-emerged and caused large outbreaks in Africa, Asia, and the Americas[10]. Notably, CHIKV arrived in the Western Hemisphere in 2013 and spread rapidly throughout the Caribbean Islands, Mexico, Central, and South America, resulting in ~3 million human infections and ~500 deaths (Source data: Pan America Health Organization)[11]. CHIKV infection in humans is biphasic, with an acute phase characteristic of active viral replication that leads to tissue damage and local inflammation, and a chronic phase typical of immunopathology. CHIKV infects many organs and cell types[10], induces apoptosis and direct tissue damage[12–14]. The acute phase is also associated with robust innate immune responses, leading to high levels of type I IFNs, proinflammatory cytokines/chemokines, growth factors[12,15–19], and immune cell infiltration[10]. In the chronic phase, CHIKV arthritis may progress without active viral replication and is characteristic of elevated cytokines and immune cell infiltration[10,20]. In mice, CHIKV infection results in a brief viremia lasting usually 5–7 days, which is initially controlled primarily by a type I IFN response[18,21–23] and is subsequently cleared by virus-specific antibody responses[24–31]. When inoculated directly into a mouse foot, CHIKV induces the first peak of foot swelling characteristic of edema at 2–3 days post infection, followed by a second peak at 6–8 days post infection[32]. Acute CHIKV infection in mice leads to infiltration of immune cells[22,33–38], among which activated macrophages are the primary cell type in infected tissues and likely a source of CHIKV persistence[36,39,40].

Autophagy, an evolutionarily conserved recycling pathway, is also an important cell-intrinsic antiviral mechanism; however, it can be hijacked by certain viruses for replication[41]. The majority of alphaviruses-related literature demonstrates an antiviral role for autophagy in both mouse and human cells[42], particularly in mouse models[43–46]; though a few in vitro studies suggest otherwise[47,48]. Here, we report that MSR1 resists CHIKV infection by inducing autophagy. $Msr1^{-/-}$ mice present increased viral loads and exacerbated foot swelling after CHIKV infection, while a normal type I IFN response. MSR1 promotes autophagy to restrict CHIKV replication by directly interacting with the core autophagy complex ATG5-ATG12. MSR1 represses CHIKV replication through the ATG5-ATG12-ATG16L1 complex, and this requires the FIP200-and-WIPI2-binding domain (FBD) that mediates autophagy, but not the WD40 domain of ATG16L1 that mediates LC3-associated phagocytosis (LAP).

## Results

**MSR1 controls CHIKV infection and pathogenesis**. Previous studies have demonstrated that the physiological functions of MSR1 during viral infections vary with viral species and disease models[4–9]. Thus, we examined the role of MSR1 in CHIKV pathogenesis in vivo. Intriguingly, $Msr1$ mRNA expression was upregulated rapidly by CHIKV in blood cells of wild-type (WT) mice, with a peak at 12–24 h post infection (p.i.). The mRNA expression of type I IFN genes, $Ifna$ and $Ifnb1$, by blood cells was gradually down-regulated after CHIKV infection, with an inverse correlation with viral loads, suggesting that CHIKV can efficiently suppresses type I IFN responses in blood cells. However, the expression of two classical interferon-stimulated genes (ISG), $Ifit1$ and $Oas1a$, in blood cells was upregulated by CHIKV, with a peak at 96 h (Fig. 1a). These observations suggest that serum IFN-I protein production activated by CHIKV is likely derived from non-hematopoietic cells, consistent with a previous study[18]. The peak induction of $Msr1$ (at 12–24 h) preceded that of viremia (at 48 h) and ISG (at 96 h), suggesting an immediate early antiviral role for Msr1. Indeed, the viremia in $Msr1^{-/-}$ mice at days 2 through 5 after infection was significantly higher than that in WT mice (Fig. 1b, c). A modest increase of viral load in $Msr1^{-/-}$ feet compared to WT was also noted at day 4 p.i. (Fig. 1d). $Msr1^{-/-}$ mice presented overt foot swelling at days 2 through 6 p.i., while WT showed a modest increase in the foot dimension (Fig. 1e). Histopathological analyses by hematoxylin and eosin staining confirmed a marked increase in immune cell infiltration into the muscles and joints of $Msr1^{-/-}$ compared to WT mice (Fig. 1f, g). The mRNA expression of $Ifnb1$, $Tnfa$, and $Ifit2$ was not impaired, with $Ifit2$ and $Ifnb1$ being modestly increased in $Msr1^{-/-}$ mice (Supplementary Fig. 1a, b). Similar results were noted from in vitro experiments; intracellular CHIKV load was increased in $Msr1^{-/-}$ when compared to WT bone marrow-derived macrophages (BMDM) (Fig. 1h). The transcript abundance of CHIKV-elicited $Ifnb1$ and $Tnfa$ was slightly higher in $Msr1^{-/-}$ than that of WT cells (Supplementary Fig. 1c). These data suggest that in vivo and in vitro MSR1 restricts CHIKV infection via an IFN-I independent, cell-intrinsic mechanism.

**MSR1 regulates autophagy during CHIKV infection**. We next attempted to pinpoint the underlying antiviral mechanism of MSR1 in macrophages. The scavenger receptors are potentially involved in autophagy induction[49,50]. We thus asked if MSR1 regulates autophagy, which has been recently shown to control CHIKV[43–46]. Of note, an autophagy-inducing peptide was demonstrated to reduce the mortality of mice infected with CHIKV[46]. Several recent publications have demonstrated the ability of CHIKV to induce autophagy[42,45,47,51]. We recapitulated this in CHIKV-infected BMDMs, one of the likely sources of CHIKV persistence in musculoskeletal tissues[36,39]. Upon autophagy activation, microtubule-associated protein 1A/1B-light

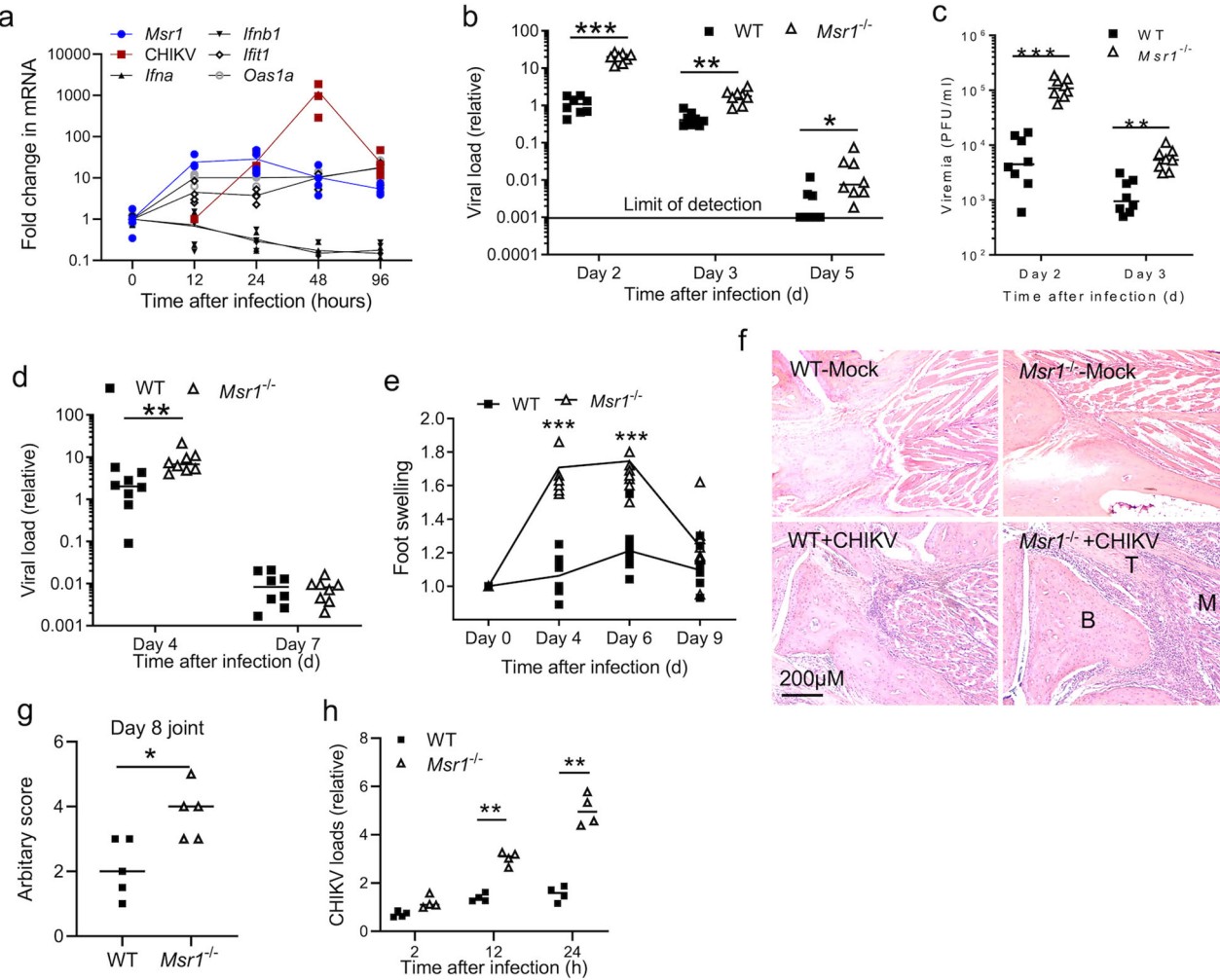

**Fig. 1 MSR1 is critical for controlling CHIKV pathogenesis in mice.** Eight-week old, sex-matched, C57BL/6 (WT) and Msr1 deficient ($Msr1^{-/-}$) mice were infected with CHIKV. Quantitative RT-PCR analyses of **a** $Msr1$, CHIKV, type I IFN, ISG mRNA ($n = 4$ mice), **b** CHIKV loads in whole blood cells of WT and $Msr1^{-/-}$ mice ($n = 8$ mice per genotype). **c** Viremia presented as plaque-forming units (PFU) per ml serum ($n = 8$ mice per genotype). **d** Quantitative RT-PCR analyses of CHIKV loads in the ankle joints ($n = 8$ mice per genotype). **e** Fold changes in the footpad dimensions of infected mice over uninfected (day 0) ($n = 8$ mice per experimental group). **f** Representative micrographs of hematoxylin and eosin staining of ankle joints at 8 days after infection. $n = 5$ mice per genotype. B: bone, T: tendon, M: muscle. Magnification: ×200. **g** Arbitrary scores of ankle joint inflammation and damage using scales of 1 to 5, with 5 representing the worst disease presentation. $n = 5$ mice per genotype. **h** Quantitative RT-PCR analyses of the cellular viral loads in bone marrow-derived macrophages (BMDM) infected with CHIKV at a multiplicity of infection (MOI) of 10, $n = 4$. Each dot=one mouse/biological repeat, the small horizontal line: the median of the result. The data represent two independent experiments. *$p < 0.05$; **$p < 0.01$; ***$p < 0.001$ [**b**-**d** non-parametric $t$-test, **e**, **g**, **h** Student's $t$-test].

chain 3 (LC3-I) is conjugated to phosphatidylethanolamine (LC3-II) of autophagosomal membranes, which is a common marker for autophagy. LC3-stained punctae were significantly increased in WT cells after CHIKV infection, but much fewer in $Msr1^{-/-}$ than WT cells (Fig. 2a, b). CHIKV-induced LC3-II protein was also reduced in $Msr1^{-/-}$ compared to WT cells (Fig. 2c). To block autophagic flux, we treated cells with chloroquine for 4 h following 12 h-and 24 h-CHIKV infection respectively. Chloroquine treatment did not change LC3-II levels in non-infected cells, but increased LC3-II levels in all infected cells, indicating induction of autophagy and enhanced autophagic flux by CHIKV. Nonetheless, there was still a decrease in LC3-II levels in $Msr1^{-/-}$ cells (Fig. 2c), suggesting that autophagy induction by CHIKV is enhanced by the presence Msr1. To see whether the antiviral function of MSR1 is evolutionarily conserved, we generated $MSR1^{-/-}$ using the CRISPR-Cas9 technology in human trophoblasts. We included $ATG12$, an essential gene for autophagy, as a positive control. The reasons for choosing trophoblasts were

that (1) human trophoblasts can maintain stable expression of MSR1 during in vitro culturing and passaging; (2) trophoblasts are permissive to CHIKV and could be physiologically relevant to CHIKV congenital transmission[52]. We confirmed successful depletion of ATG12 and MSR1[53] protein expression by Western blotting (Fig. 2d), and observed increased CHIKV replication in $MSR1^{-/-}$ and $ATG12^{-/-}$ cells (Fig. 2e). CHIKV infection resulted in LC3-II conversion in wild-type cells. In contrast, LC3-II was completely absent from $ATG12^{-/-}$ or reduced in $MSR1^{-/-}$ cells (Fig. 2f). To validate the deficiency in autophagosome formation in $MSR1^{-/-}$ cells during CHIKV infection, we next used DAPGreen, a fluorescent dye that is incorporated into autophagosomal membranes. Consistent with LC3-staining and immunoblotting results, DAPGreen punctae were fewer in $MSR1^{-/-}$ than WT cells (Fig. 2g). Furthermore, LC3-I to LC3-II conversion was induced in WT mouse feet after CHIKV infection and the LC3-II levels were lower in $Msr1^{-/-}$ than WT mice (Fig. 2h). LC3 lipidation could be driven by the canonical, non-canonical

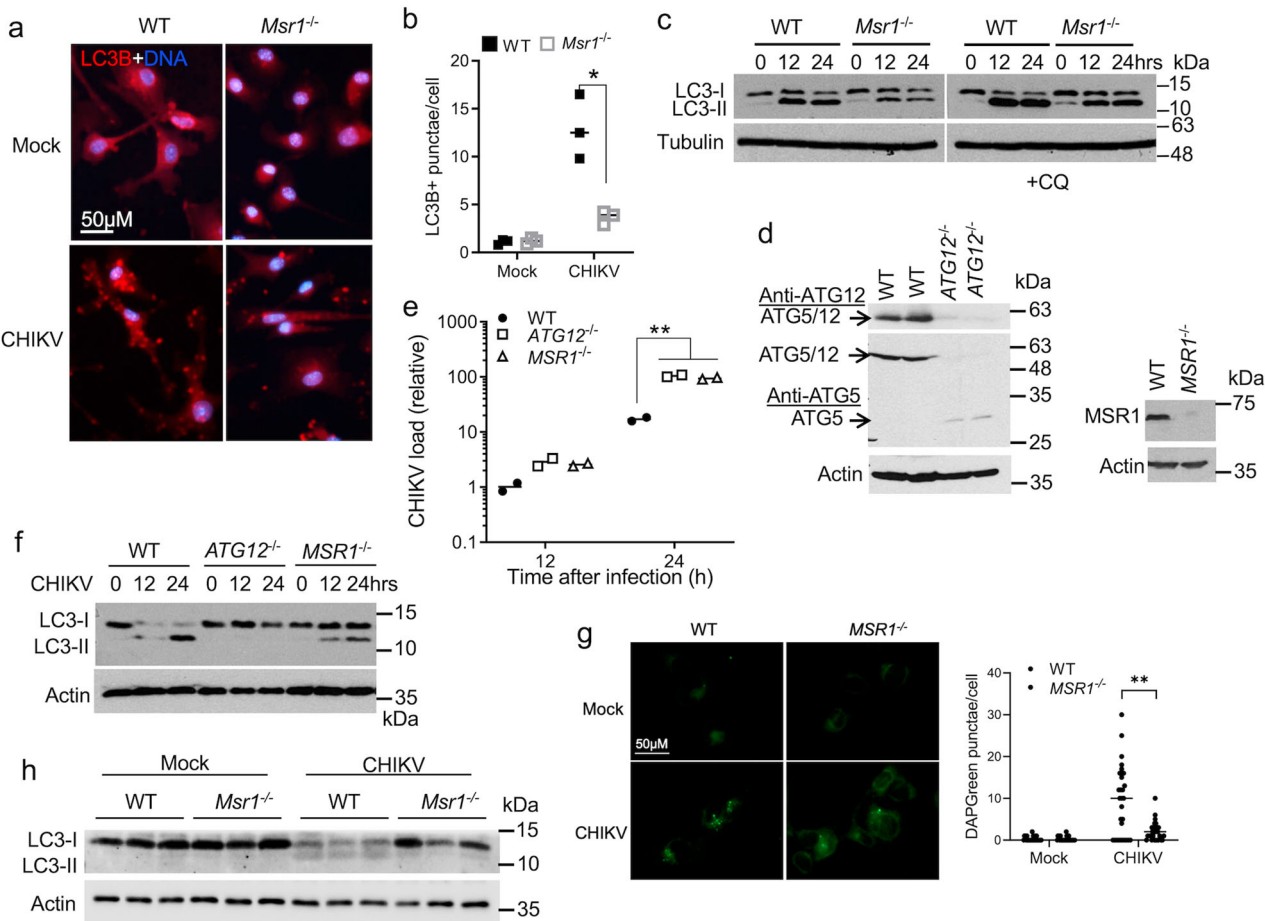

**Fig. 2 MSR1 is critical for CHIKV-induced autophagy. a** Microscopic images of immunofluorescence staining for LC3B in BMDMs infected with CHIKV at a MOI of 10 for 12 h. **b** The mean number of LC3B-positive punctae per cell ($n = 30$ cells per field × 3 fields). *$p < 0.05$ (two-tailed Student's $t$-test). **c** Immunoblots of LC3B in BMDMs infected with CHIKV at a MOI of 10, following treatment without/with 20 μM of chloroquine (CQ) for 3 h. **d** Generation of human $ATG12^{-/-}$ and $MSR1^{-/-}$ trophoblasts by CRISPR-Cas9. Both anti-ATG12 and ATG5 antibodies detect the stable ATG5-ATG12 dimer, which is abolished in $ATG12^{-/-}$ cells. Monomeric ATG5 is liberated and detected in $ATG12^{-/-}$ cells by an anti-ATG5 antibody. **e** Quantitative RT-PCR analyses of intracellular CHIKV RNA in trophoblasts infected with CHIKV at a MOI of 1, $n = 2$ biologically independent samples, **$p < 0.01$ (two-way ANOVA). **f** Immunoblots of LC3B in trophoblasts infected with CHIKV at a MOI of 1. **g** Microscopic images and quantification of DAPGreen punctae (per cell, $n = 30$ cells) in live trophoblasts infected with CHIKV at a MOI of 1 for 10 h. **$p < 0.01$ (two-tailed Student's $t$-test). **h** Immunoblots of LC3B in mouse feet mock-treated or infected with CHIKV for 4 days. Each lane represents one animal. The uncropped immunoblots for all figures can be found in Supplementary Fig. 3.

autophagy and autophagy-unrelated ATG5-ATG12-ATG16L1 function[54]. SQSTM1/p62 is a cargo protein that targets ubiquitinated proteins to autophagosome for degradation. We next examined the p62 degradation. In contrast to our expectations and a recent study in human cells[47], the p62 protein level was highly increased in BMDMs after CHIKV infection. We did not see a difference in the p62 protein level between WT and $Msr1^{-/-}$ cells (Supplementary Fig. 2a). However, p62 aggregation was observed after CHIKV infection (Supplementary Fig. 2b), suggesting induction of autophagy. Then we asked if the increase in p62 protein level could be due to the induction of de novo synthesis of p62 by CHIKV infection. Indeed, the RT-PCR results reveal that the p62 mRNA was induced by CHIKV infection as early as 4 h after infection (Supplementary Fig. 2c, d). These data suggest that the rapid de novo p62 synthesis may override autophagic degradation.

**MSR1 interacts with ATG5-ATG12 complex following CHIKV infection**. Given that the components involved in autophagy formation have been well established, we therefore performed a small scale, candidate-based co-immunoprecipitation screening

to identify MSR1-interacting autophagy-related proteins involved in initiation (ULK1, ATG9A, ATG13), nucleation (Beclin 1, VPS34), and expansion (ATG5-ATG12, ATG16L1). To this end, FLAG-tagged autophagy genes of mouse origin and Myc-Msr1 were co-expressed in HEK293T cells and subjected to immunoprecipitation with an anti-FLAG antibody. Intriguingly, Msr1 bound both Atg5 and Atg12, preferably Atg12 (Fig. 3a). This is not surprising since ATG5 and ATG12 always exist in a heterodimer complex. The binding of recombinant FLAG-Msr1 to endogenous Atg5/12 complex was enhanced at 12 and 24 h after CHIKV infection compared to 0 h (no virus) (Fig. 3b). Since any a protein, when overexpressed, could bind another protein artificially, we thus examined endogenous Msr1-Atg12 interaction. To this end, we immunoprecipitated Msr1 and its binding proteins using an anti-Msr1 antibody. Msr1 co-precipitated with Atg5-Atg12 complex at 12 and 24 h after CHIKV infection in BMDMs (Fig. 3c). In uninfected cells, Msr1 was predominantly localized to the plasma membrane and Atg12 was diffuse in the cytoplasm. After CHIKV infection, Atg12 and Msr1 co-localized to large intracellular punctae (Fig. 3d). To further validate MSR-ATG12 interaction, we next pinpointed the functional domain

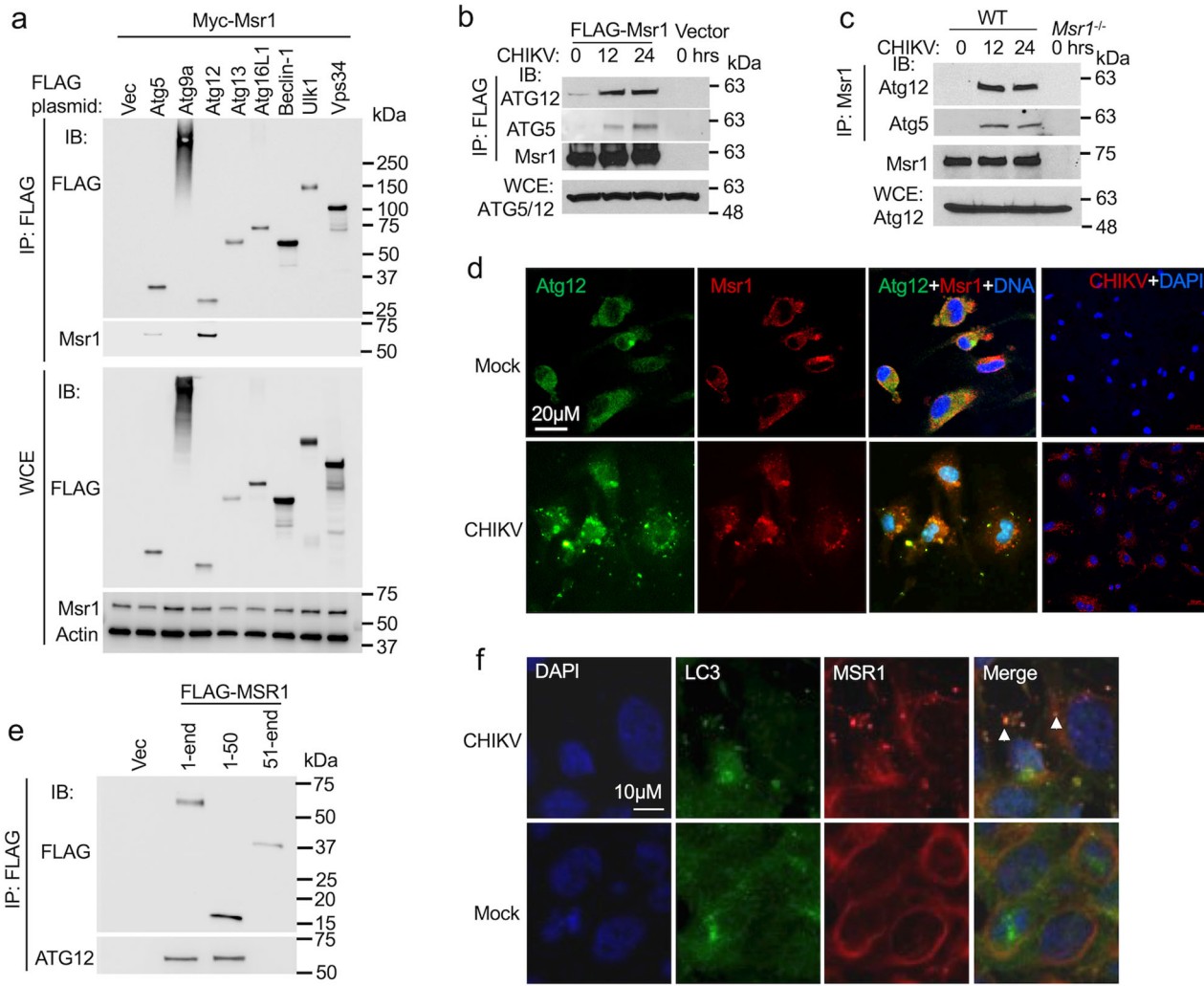

**Fig. 3 CHIKV infection induces MSR1 interaction with the ATG5-ATG12 complex. a** Co-immunoprecipitation (co-IP) of FLAG-tagged mouse autophagy proteins with Myc-Msr1 expressed in HEK293 cells using a mouse monoclonal anti-FLAG antibody, followed by immunoblotting (IB) with a mouse monoclonal anti-FLAG and rabbit anti-Msr1 antibody. **b** Co-IP of FLAG-Msr1 (mouse) with endogenous human ATG5-ATG12 complex expressed in HEK293. CHIKV infection: MOI of 1. WCE: whole-cell extract. **c** Co-IP of endogenous Msr1 with the Atg5-Atg12 complex from bone marrow-derived macrophages (BMDMs) following CHIKV infection at a MOI of 10 for 12 and 24 h. The IP was carried out with a rabbit anti-Msr1 antibody cross-linked to protein A/G agarose beads. $Msr1^{-/-}$ cell serves as a negative control. **d** Immunofluorescence staining for Atg12 and Msr1 in BMDMs infected without (mock) /with CHIKV at a MOI of 10 for 12 h. Atg12 and Msr1 were stained by a mouse anti-Atg12 and rabbit anti-Msr1, followed by an Alexa Fluor-488 (green) and -594 (red)-conjugated secondary antibody respectively. The cells were counterstained for nuclei by DAPI (blue). The yellow punctae in the overlay indicate co-localizations. Magnification: ×400. **e** Co-IP of FLAG-MSR1 (human) or its fragment (aa1-50, 51-end) with endogenous ATG5-ATG12 complex expressed in HEK293 cells. CHIKV infection: MOI of 1 for 12 h. **f** Microscopic images of immunofluorescence staining for LC3B and MSR1 in trophoblasts without (mock)/infected with CHIKV at a MOI of 0.5 for 12 h. The yellow punctae in the overlay indicate co-localizations. Magnification: ×400. The uncropped immunoblots for all figures can be found in Supplementary Fig. 3.

within MSR1 that directly interacts with ATG12. MSR1 has a small cytoplasmic N-tail that can interact with intracellular proteins such as TRAF6[55]. Thus, we postulate that MSR1 uses its cytoplasmic tail to interact with ATG12. We truncated human MSR1 into two fragments, the cytoplasmic N-tail (amino residues 1–50) and the extracellular C-terminal part with a transmembrane domain (amino residues 51-end). Indeed, both full-length and the cytoplasmic tail co-precipitated with endogenous ATG5-ATG12 complex, but the extracellular 51-end fragment failed to do so (Fig. 3e). Moreover, immunofluorescence microscopy shows that endogenous human MSR1 was localized to LC3-positive punctae after CHIKV infection in trophoblasts (Fig. 3f). Altogether, the above-mentioned data demonstrate that MSR1 interacts with ATG5-ATG12 upon CHIKV infection and this is evolutionarily conserved.

**MSR1 inhibits CHIKV infection via ATG5-ATG12-ATG16L1.** To establish a functional link between MSR1 and ATG5-ATG12, we first examined the antiviral activity of mouse Msr1 over-expression in Atg5-deficient mouse embryonic fibroblasts (MEF). FLAG-tagged mouse Msr1 expression was confirmed by immunoblotting, with comparable levels between $Atg5^{-/-}$ and $Atg5^{+/+}$ cells (Fig. 4a). CHIKV replication was enhanced in $Atg5^{-/-}$ cells compared to $Atg5^{+/+}$ cells ($Atg5^{-/-}$ +Vec v.s $Atg5^{+/+}$ +Vec) (Fig. 4b). Overexpression of Msr1 in $Atg5^{+/+}$ cells repressed CHIKV replication in a dose-dependent manner ($Atg5^{+/+}$ +Msr1 v.s $Atg5^{+/+}$ +Vec, solid lines), but this effect was not seen in $Atg5^{-/-}$ cells ($Atg5^{-/-}$ +Msr1 v.s $Atg5^{-/-}$ +Vec, broken lines) (Fig. 4b). We next further validate the functional link between MSR1 and ATG5-ATG12 in human cells. Over-expression of human MSR1 reduced CHIKV replication in WT

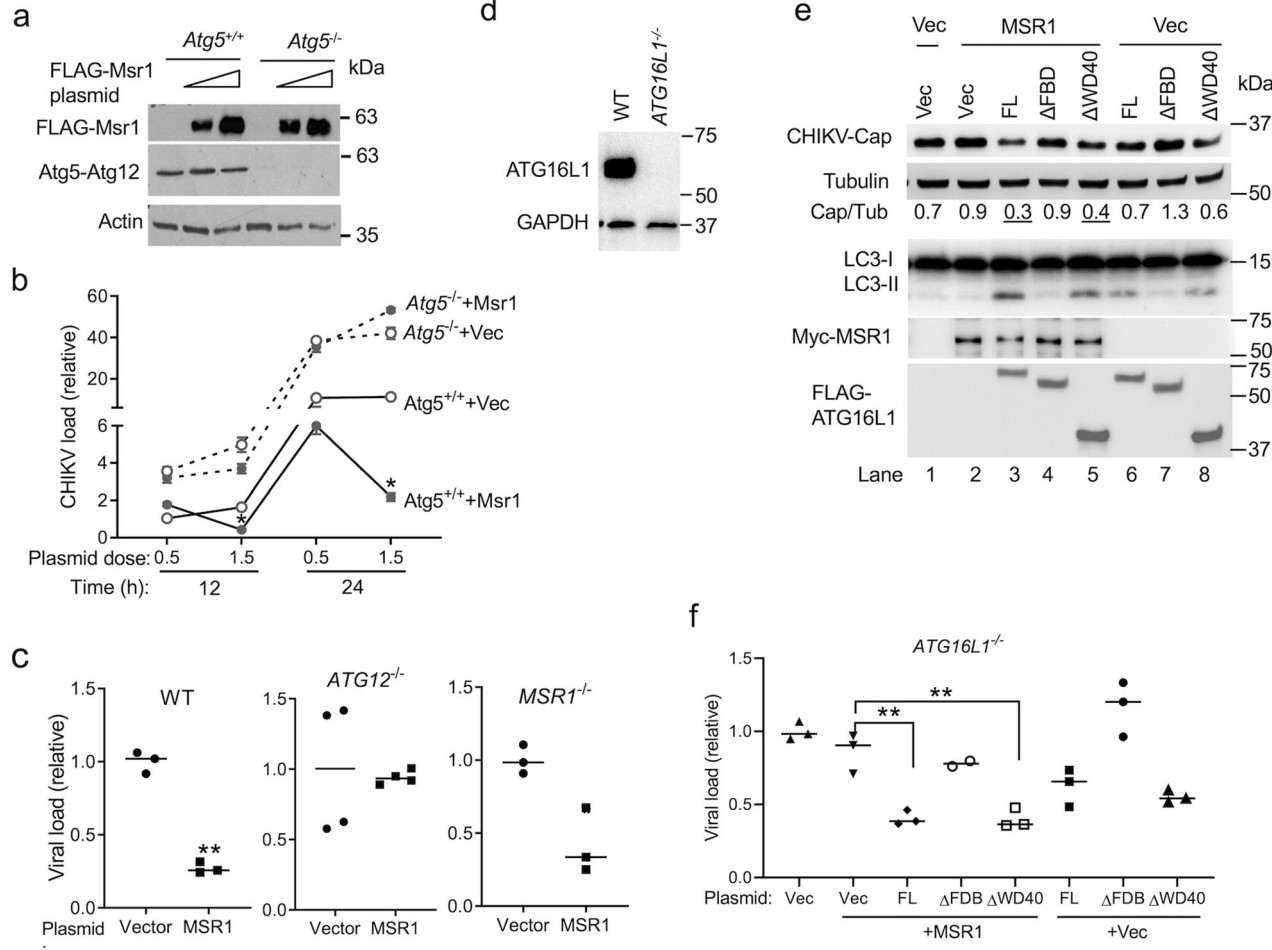

**Fig. 4 Inhibition of CHIKV replication by MSR1 is dependent on ATG5-ATG12-ATG16L1. a, b** Repression of CHIKV replication by MSR1 requires ATG5. Mouse primary embryonic fibroblasts were transfected with either 0.2 or 1.5 µg of empty vector (Vec) or FLAG-Msr1 (mouse) expression plasmid. Twenty-four hours later the cells were then infected with CHIKV (multiplicity of infection MOI = 0.5). **a** Immunoblots showing FLAG-Msr1 (mouse) and Atg5-Atg12 expression at 24 h after plasmid DNA transfection. **b** Quantitative RT-PCR analyses of intracellular CHIKV RNA. $N = 2$. **c** Repression of CHIKV replication by MSR1 requires ATG12. Human trophoblasts were transfected with either 0.2 µg of empty vector or a FLAG-MSR1 (human) expression plasmid. Twenty-four hours later, the cells were then infected with CHIKV MOI = 0.5 for 12 h. Intracellular CHIKV RNA was quantitated by RT-PCR. $n = 3$ biologically independent samples. **d** Generation of human $ATG16L1^{-/-}$ trophoblasts by CRISPR-Cas9. The immunoblot shows $ATG16L1$ knockout efficiency. **e** Repression of CHIKV replication by MSR1 requires the FBD domain of ATG16L1. $ATG16L1^{-/-}$ trophoblasts were transfected with the indicated combinations of expression plasmids (human gene) and vector (Vec) (50 ng each) respectively. After 24 h, the cells were infected with CHIKV at a MOI of 0.5 for 16 h. FL: full-length, ΔFBD: FBD domain deletion, ΔWD40: WD40 domain deletion of ATG16L1. The immunoblots show CHIKV Capsid and cellular protein expression. $n = 3$ biologically independent samples. The small horizontal line: the median of the result. $*p < 0.05$, $**p < 0.01$ (Student's $t$-test). The uncropped immunoblots for all figures can be found in Supplementary Fig. 3.

and $MSR1^{-/-}$ cells, but not in $ATG12^{-/-}$ cells (Fig. 4c). These data suggest that the anti-CHIKV activity of MSR1 is dependent on ATG5-ATG12 complex.

In order to further clarify the role of MSR1 in autophagy, we then dissected the functional domains of ATG16L1 in MSR1-mediated antiviral mechanism. ATG16L1 forms a complex with ATG5-ATG12 to activate LC3 lipidation during both canonical and non-canonical autophagy. The C-terminal WD40 domain (amino residues 336–632) of ATG16L1 is essential for LAP, but dispensable for ULK1/VPS34-mediated canonical autophagy[54]. The FBD domain (amino residues 219–242) contains the binding sites for FIP200/ WIPI2 and is required for ULK1-mediated autophagy. To this end, we generated $ATG16L1^{-/-}$ trophoblast cells (Fig. 4d) and reconstituted them with human full-length, ΔFBD or ΔWD40 of ATG16L1 gene respectively. We then examined the effect of MSR1 overexpression on CHIKV replication and LC3 lipidation in these cells. Compared to the vector control, ectopic expression of

full-length MSR1 enhanced LC3-II and inhibited viral capsid expression in WT- (Lane 3 versus Lane 6) and ΔWD40- (Lane 5 versus Lane 8), but not ΔFBD-(Lane 4 versus Lane 7) reconstituted cells (Fig. 4e). In agreement with the immunoblotting results for CHIKV capsid, quantitative RT-PCR analyses show that viral load was decreased most significantly when MSR1 was overexpressed in full-length ATG16L1- or ΔWD40-reconstituted cells (Fig. 4f). These results demonstrate that the antiviral function of MSR1 requires the FBD, but not WD40. Altogether, our results suggest that MSR1 activates ATG5-ATG12-ATG16L1-dependent autophagy, but not LAP, to control CHIKV infection.

**CHIKV nsP1 interacts with MSR1 and enhances MSR1-ATG12 interaction.** To understand how MSR1 is activated by CHIKV, we examined if any CHIKV proteins directly interact with MSR1. We co-transfected FLAG-MSR1 and individual Myc-tagged CHIKV

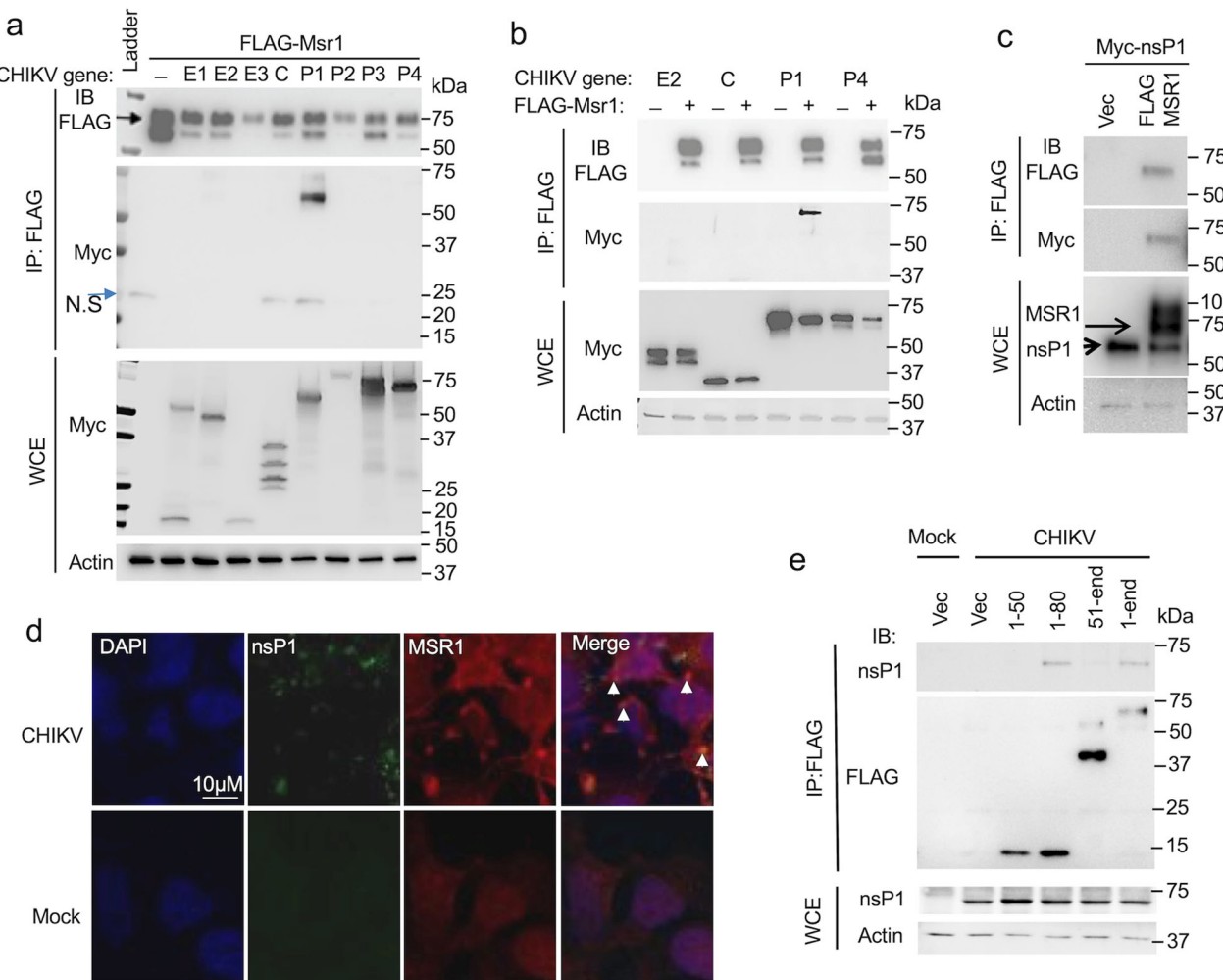

**Fig. 5 MSR1 interacts with CHKV nsP1. a**, **b** Co-immunoprecipitation (co-IP) of FLAG-Msr1 (mouse) with Myc-tagged CHIKV proteins expressed in HEK293 cells using a mouse anti-FLAG antibody (IP), followed by immunoblotting (IB) with a rabbit anti-FLAG and anti-Myc antibody. WCE: whole-cell extract. N.S: non-specific bands. E: CHIKV envelope, C: capsid, P: non-structural protein. **c** co-IP of FLAG-MSR1 (human) with Myc-nsP1 in HEK293. **d** Immunofluorescence staining for endogenous MSR1 and CHIKV nsP1 in trophoblasts without (mock) / infected with CHIKV at a multiplicity of infection of 0.5 for 12 h. MSR1 and nsP1 were stained by a rabbit anti-MSR1 and rat anti-nsP1, followed by an Alexa Fluor-594 (red) and -488 (green) conjugated secondary antibody respectively. The cells were counterstained for nuclei by DAPI (blue). Scale bar: 20 μM. The yellow punctae in the overlay indicate co-localizations. **e** Co-IP of FLAG-MSR1 fragments with endogenous nsP1 in trophoblasts infected with CHIKV (MOI = 0.5 for 12 h) using a mouse anti-FLAG antibody (IP), followed by IB with a rabbit anti-FLAG and anti-nsP1 antibody. Vec: vector. Mock: mock infection. The uncropped immunoblots for all figures can be found in Supplementary Fig. 3.

gene plasmids into HEK293T cells, and immuno-precipitated FLAG-MSR1 with an anti-FLAG antibody. The results show that both human and mouse MSR1 co-precipitated specifically with Myc-nsP1 (Fig. 5a–c). Immunofluorescence staining demonstrates that viral nsP1 co-localized with endogenous MSR1 under physiological conditions (Fig. 5d). To further strengthen nsP1-MSR1 interaction, we examined the specific domain of MSR1 that mediates MSR1-nsP1 interaction. While 1-50aa and 51-end of MSR1 bound endogenous nsP1 very weakly, 1-80aa with the transmembrane domain bound nsP1 much better (Fig. 5e), suggesting that MSR1 interaction with nsP1 requires both 1-50aa and the membrane anchorage of MSR1. Intriguingly, MSR1-ATG12 interaction was enhanced by the presence of nsP1 (Fig. 6a). We then asked if nsP1 forms a complex with ATG12 and MSR1. Indeed, FLAG-ATG12 co-immunoprecipitated with Myc-nsP1 and endogenous nsP1 during CHIKV infection; and this interaction was enhanced by overexpression of MSR1 (Fig. 6b, c). While nsP1 localization to LC3-positive autophagosomes was decreased in $MSR1^{-/-}$ when

compared WT cells (Fig. 6d). We then asked if nsP1 is brought by MSR1 to ATG5-ATG12 for autophagic degradation. To this end, we used rapamycin to trigger canonical autophagy and included Capsid which is targeted by SQSTM1/p62 to autophagolysosomes for degradation[47]. Myc-Capsid protein was reduced in the presence of rapamycin and this degradation was blocked by chloroquine, an autophagolysomal inhibitor. However, the nsP1 protein level was not altered after rapamycin treatment (Fig. 6e). We then asked if CHIKV infection triggers pathways other than autophagy to degrade nsP1. To this end we transfected Myc-nsP1 and Capsid expression plasmids into WT and $P62^{-/-}$ trophoblasts (Fig. 6f, g) and infected cells with a low MOI of CHIKV for 12 h. The Myc-nsP1 level was not reduced after CHIKV infection compared to before infection in both WT and knockout cells; Myc-Capsid was degraded in CHIKV-infected WT cells, but not in $P62^{-/-}$ (Fig. 6g). These results suggest that binding of nsP1 to MSR1 enhances MSR1-ATG12 interaction and thus autophagic degradation of Capsid.

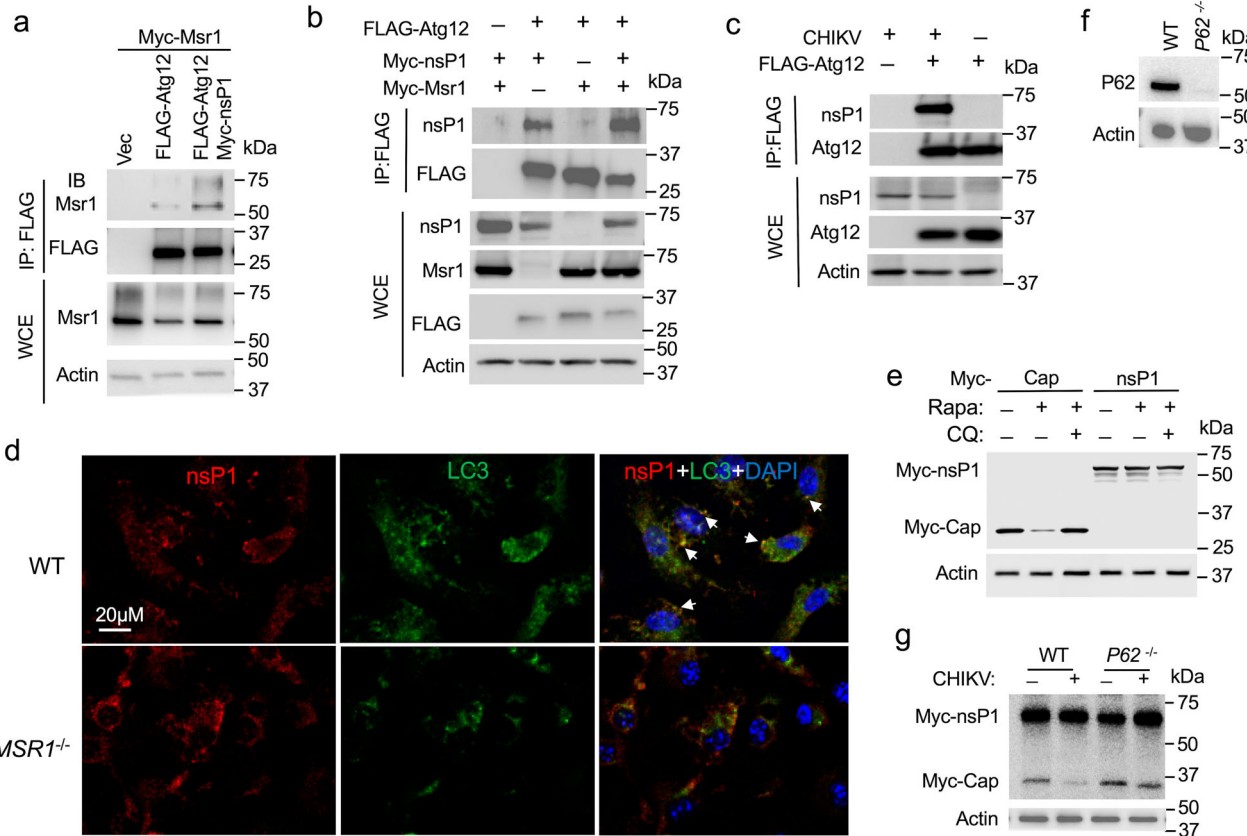

**Fig. 6 CHIKV nsP1 enhances MSR1-ATG12 interaction. a** Co-immunoprecipitation (co-IP) of FLAG-Atg12 with Myc-Msr1 (mouse) proteins in the presence of nsP1 in HEK293T cells using a mouse monoclonal anti-FLAG antibody, followed by immunoblotting (IB) with a rabbit anti-FLAG or Msr1 antibody. WCE: whole-cell extract. **b** Co-IP of FLAG-Atg12 with Myc-nsP1 in the presence of Myc-Msr1 in HEK293T cells using a mouse monoclonal anti-FLAG antibody, followed by IB with a rabbit anti-FLAG, nsP1 or Msr1 antibody. **c** co-IP of FLAG-Atg12 with endogenous ns1 in trophoblasts infected with CHIKV at a multiplicity of infection (MOI) of 0.5 for 12 h using a mouse anti-FLAG antibody (IP), followed by IB with a rabbit anti-FLAG and anti-nsP1 antibody. **d** Immunofluorescence staining for endogenous CHIKV nsP1 and LC3 in trophoblasts infected with CHIKV at a MOI of 0.5 for 12 h. The cells were counterstained for nuclei by DAPI (blue). The arrows indicate co-localizations. **e** Immunoblots of Myc-tagged CHIKV proteins after rapamycin (Rapa) treatment. Myc-tagged CHIKV gene expression plasmids were transfected into HEK293T cells for 24 h. The cells were then treated with 100 nM rapamycin with/without 40 μM of chloroquine (CQ) for 24 h. **f** The immunoblots show P62 knockout by CRISPR-Cas9 in human trophoblasts. **g** Immunoblots of Myc-tagged CHIKV proteins in trophoblasts after CHIKV infection. Myc-Cap, nsP1 expression plasmids were transfected into trophoblasts for 24 h. The cells were then infected with CHIKV at a MOI of 0.5 for 12 h. The uncropped immunoblots for all figures can be found in Supplementary Fig. 3.

## Discussion

Although MSR1 has been implicated in recognition of viral nucleic acids, activation of IFN-I[4,5], macrophage adherence[6], M2 macrophage polarization[7], and activation of neutrophil-mediated complement[8], its molecular mechanism of action remains elusive. In the current study, we demonstrate a critical role for MSR1 in controlling CHIKV pathogenesis and provide several lines of evidence showing how MSR1 participates in autophagy. Without Msr1, mice are more prone to CHIKV-elicited arthritic pathologies in feet (Fig. 1e–g), likely a result of failure to control viral replication (Fig. 1b–d), which induces apoptosis and direct tissue damage[12–14]. Moreover, immune cell infiltration is a hallmark of CHIKV-elicited pathologies, with monocytes and macrophages being a dominant component and likely a source of CHIKV persistence[36,39,40]. Considering a putative role for MSR1 in promoting M2 macrophage polarization (anti-inflammatory) in virus-elicited hepatitis[7], it is thus plausible that MSR1 could limit CHIKV arthritis pathogenesis via M2, in addition to controlling viral replication through autophagy.

Type I IFNs are critical for limiting early CHIKV infection in vivo[18,21–23], and MSR1 has been shown to sense extracellular viral nucleic acids, mediating their endocytosis and presenting them to TLR3/9, thereby triggering a IFN-I response in both

infected and uninfected cells[4,5]. However, we observed no deficiency in IFN-I response in vivo and in vitro. This discrepancy could be due to viral species and different PRR pathways that different viruses activate. For instance, although both are RNA viruses, encephalomyocarditis virus activates MDA5 and vesicular stomatitis virus triggers RIG-I-dependent IFN-I responses respectively[56]. CHIKV-induced type I IFN responses are predominantly dependent on RIG-I and MDA5[57]. In favor of our results, a previous study showed that ectopic expression of MSR1 inhibited Sindbis virus (an alphavirus) and human parainfluenza virus type 3 in $STAT1^{-/-}$ fibroblasts[9], suggesting an IFN-I-independent antiviral function for MSR1. Moreover, MSR1 expression was dramatically upregulated in mice at the very early stage of CHIKV infection (12 h after infection), way before the peak of viremia (48 h) and ISG expression (96 h). These observations further suggest an early IFN-I-independent antiviral role for MSR1. MSR1 could also regulate macrophage adherence and help clear microbial pathogens in vivo[6]. However, we found no deficiency in adherence of bone marrow-derived $Msr1^{-/-}$ macrophages to tissue-culture treated plastic surfaces. A plausible explanation may be the source of macrophages, which are derived from bone-marrow in our case and from thioglycollate-treated peritonea in Suzuki's study[6]. Distinct from the published results,

our study provides several pieces of evidence for a cell-intrinsic antiviral mechanism of MSR1. We demonstrate here that MSR1 controls CHIKV infection via ATG5-ATG12-ATG16L1, the core components of autophagy and LAP. This is evidenced by CHIKV-induced interaction between MSR1 and ATG5-ATG12 complex (Fig. 3). This interaction is mediated by the cytoplasmic tail of MSR1 (Fig. 3e) and probably requires MSR1 trafficking from the plasma membrane to internal membrane structures[55]. Repression of CHIKV replication by MSR1 depends on ATG5-ATG12 and the FBD domain (mediating autophagy) of ATG16L1, but not the WD40 domain (mediating LAP) of ATG16L1 (Fig. 4). These data suggest that MSR1 participates in ATG5-ATG12-ATG16L1-depedent autophagy.

Alphavirus replication takes place within large cytoplasmic vacuoles (CPV), derivatives of late endosomes and lysosomes, the inner surface of which are covered by small spherules or the viral replication complexes. Replication complexes arise by assembly of the four viral non-structural proteins (nsP) and genome at the plasma membrane, guided there by nsP1[58,59]. Intriguingly, our results demonstrate that MSR1 interacts specifically with nsP1 and this interaction does not direct nsP1 to autophagolysomal degradation (Figs. 5, 6). Instead, nsP1 seems to trigger MSR1-ATG12 interaction (Fig. 6a), thus promoting ATG5-ATG12-ATG16L1-autophagy to degrade CHIKV Capsid protein. It is also plausible that MSR1 could interfere with CHIKV replication complexes through ATG5-ATG12-ATG16L1-mediated lipidation and attachment of LC3 to non-autophagosomal membranes[60,61]. Indeed, MSR1 localizes to LC3-postive membrane structures after infection (Fig. 3f) and non-autophagosomal membrane-associated LC3 could restrict RNA virus replication[60,61].

In summary, in this study we provide in vivo and in vitro evidence for a critical anti-CHIKV role of MSR1-ATG12 axis. This pathway may be also applicable to other alphaviruses as evidenced by a recent study[9]. Future efforts will be focused on the elucidation of the common mechanism by which all alphaviruses activate MSR1-ATG12 axis.

## Methods

**Mouse models**. All mice were obtained from the Jackson Laboratory and had the same genetic background (C57BL/6) and housing conditions. The control mice were C57BL/6. $Msr1^{-/-}$ mice[6] were made from 129 embryonic stem (ES) cells and then backcrossed to C57BL/6 mice for 12 generations (https://www.jax.org/strain/006096). These gene-deficient mice were normal in body weight and general health, when compared to C57BL/6. We used 8–12 weeks old sex-matched mice for all the experiments. All animal protocols were approved by the Institutional Animal Care & Use Committee at Yale University and New York Medical College adhering to the National Institutes of Health recommendations for the care and use of laboratory animals.

**Reagents and antibodies**. The rabbit anti LC3B (Cat # 2775), GAPDH (Cat #5174), Tubulin (Cat# 2148), ATG5 (Cat#12994), Actin (Cat # 8456) and anti-p62/SQSTM11(Cat# 5114, #7695) antibodies were purchased from Cell Signaling Technology (Danvers, MA 01923, USA). The mouse anti-FLAG (Cat# TA50011) and rabbit anti-human/mouse MSR1 (Cat# TA336699) antibodies were from Origene (Rockville, MD 20850, USA); the goat anti-mouse MSR1 (Cat# AF1797), mouse ant-human MSR1 (Cat# MAB2708), mouse anti-ATG12 (Cat# MAB6807) and recombinant mouse IFN-α (Cat #:12100-1) from R&D Systems (Minneapolis, MN 55413, USA). The mouse anti-CHIKV (Clone A54Q, Cat# MA5-18181) and Lipofectamine 2000 were obtained from ThermoFisher Scientific (Rockford, IL 61105, USA). The rat anti-CHIKV nsP1 (Cat# 111441) and nsP2 (Cat# 111442) were available from Antibody Research Corporation (St Peters, MO 63304, USA) and rabbit anti-CHIKV nsP1 (Cat# 11-13020) from ABGENEX (Bhubaneswar, Odisha 751024, India). DAPGreen (Cat# D676) was purchased from Dojindo Molecular Technologies, Inc. (Rockville, MD 20850).

**Plasmid construction**. The Myc-DDK-tagged mouse Msr1 ORF clone (Cat# MR205384), Atg12 (Cat# MR200886), Atg5 (Cat# MR203691), Atg16l1 (Cat# MR209513), Atg9a (Cat# MR208753), Atg13 (Cat# MR207683) and, p62/Sqstm1 (Cat# MR226105), human MSR1 (Cat# RC209609) and Beclin 1 (Cat# MR207162) were obtained from Origene (Rockville, MD 20850, USA). pcDNA-FLAG-Ulk1 (Plasmid # 27636)[62] and pcDNA4-VPS34-FLAG (Plasmid # 24398)[63] were

purchased from Addgene (Cambridge, MA 02139, USA). Myc-Msr1 was constructed by inserting Msr1 ORF (amplified from Origene clone MR205384) into the pcDNA3.0-Myc vector. The LR2006-OPY1 CHIKV genes were amplified by PCR and subcloned into pcDNA3.0-Myc. Human ATG16L1, ATG16L1ΔFBD (amino residues 229–242 deleted) and ATG16L1ΔWD40 (amino residues 337-end) amplified by PCR and inserted into pcDNA3.1 (Zeo)-FLAG vector. The primers are listed in Supplementary Table 1.

**Cell culture and viruses**. Human embryonic kidney 293 cells transformed with T antigen of SV40 (HEK293T, # CRL-3216), Vero cells (monkey kidney epithelial cells, # CCL-81), immortalized human trophoblasts HTR-8/SVneo (#CRL-3271) and L929 (mouse fibroblast cells, # CCL-1) were purchased from American Type Culture Collection (ATCC) (Manassas, VA20110, USA). HEK293T and Vero cells were grown in DMEM (Life Technologies, Grand Island, NY 14072 USA) supplemented with 10% FBS and antibiotics/antimycotics. These cell lines are not listed in the database of commonly misidentified cell lines maintained by ICLAC, and not authenticated or tested for mycoplasma contamination in our hands. In order to ensure cell culture mycoplasma free, we regularly treated cells with MycoZap (Lonza). For the preparation of MEF, pregnant females were euthanized on day 14 of gestation. Heterologous and homogenous fibroblasts were isolated from the same littermates (pooled from three mice per genotype). Embryos were decapitated and eviscerated then digested with trypsin for 10 min at 37 °C rotating. Fibroblasts were filtered through 100 μM filters, cultured in RPMI1640 medium (Life Technologies, NY 14072 USA) supplemented with 10% fetal bovine serum and antibiotics/antimycotics, propagated for two passages and then frozen[64]. Bone marrows were isolated from the tibia and femur bones and then differentiated into macrophages (BM-Ms) in L929 conditioned medium (RPMI1640, 20%FBS, 30% L929 culture medium, antibiotics/antimycotics) in 10-cm Petri-dishes at 37 °C for 5 days. The attached BM-Ms were cultured in tissue-culture treated plastic wares in regular RPMI1640 medium overnight before further treatment.

The CHIKV French La Reunion strain LR2006-OPY1 was a kind gift of the Connecticut Agricultural Experiment Station at New Haven, CT, USA. All viruses were propagated in Vero cells. Unless specified, MEFs were infected with CHIKV at a multiplication of infection (MOI) of 0.3, trophoblasts at a MOI of 1, BM-Ms at a MOI of 10.

**Generation of human gene knockout cells with CRISPR-Cas9 technology**. The gene-specific guide RNA was cloned into lentiCRISPRv2 vector and co-transfected into HEK293T cells with the packaging plasmids pVSVg and psPAX2[65,66]. Forty-eight hours after transfection the lentiviral particles in the cell culture media were applied to cells for 48 h. The transduced cells were then selected with puromycin at 1 μg/ml for 48–72 h. Successful knockout clones were confirmed by immunoblotting. The guide RNAs are listed in Supplementary Table 1. The wild-type control was lentiCRISPRv2 vector only.

**Mouse infection and disease monitoring**. CHIKV was propagated in Vero cells to a titer of ~$10^7$ plaque-forming units (PFU)/ml and prepared in 50 μl of sterile 1% gelatin in phosphate-buffered saline (PBS) for injection. In brief, age-and sex-matched wild-type and specific gene deleted mice (>8 weeks old) were infected with $5 \times 10^4$ PFU of La Reunion strain LR2006-OPY1 at the ventral side of a hind foot subcutaneously.

Foot swelling as an indicator of local inflammation was recorded over a period of 9 days after infection. The thickness and width of the perimetatarsal area of the hind feet were measured using a precision metric caliper in a blinded fashion. The foot dimension was calculated as width × thickness, and the results were expressed as fold increase in the foot dimension after infection compared to before infection (day 0 baseline). For histology, footpads/joints were fixed in 4% paraformaldehyde (PFA), decalcified, and processed for hematoxylin and eosin staining. Arbitrary arthritic disease scores (on a 1–5 scale with 1 being the slightest, 5 the worst) were assessed using a combination of histological parameters, including exudation of fibrin and inflammatory cells into the joints, alteration in the thickness of tendons or ligament sheaths, and hypertrophy and hyperlexia of the synovium[67] in a double-blinded manner by a trained pathologist at the NYMC histology core.

**Reverse transcription and quantitative RT-PCR**. Animal tissues/25 μl of whole blood were collected in 350 μl of RLT buffer (QIAGEN RNeasy mini kit). Soft tissues were homogenized using an electric pellet pestle (Kimble Chase LLC, USA). For ankle joints, skin and muscles were first removed, and the joints were then chopped into small pieces with a scissor in RLT buffer. RNA was extracted following the QIAGEN RNeasy protocol and, reverse-transcribed into cDNA using the BIO-RAD iScript™ cDNA Synthesis Kit. Quantitative RT-PCR was performed with gene-specific primers and 6FAM-TAMRA (6-carboxyfluorescein–N,N,N,N-tetramethyl-6-carboxyrhodamine) probes or SYBR Green. Results were calculated using the $-\Delta\Delta$Ct method and beta-actin gene as an internal control. The qPCR primers and probes for immune genes were reported in our previous studies[64,68]. The Taqman gene expression assays for Ifit1 (Mm00515153_m1), Oas1a (Mm00836412_m1), Isg15 (Mm01705338_s1), Ifit2 (Mm00492606_m1), and Cxcl10 (Mm00445235_m1) were obtained from ThermoFisher Scientific. The other qPCR primers are summarized in Supplementary Table 1.

**Virus titration**. Plaque-forming assays with tissues, cell culture medium or plasma were performed as previously described[69]. In brief, 100 µl of samples diluted with sterile PBS by 10–100 folds, or 30–100 µg (total proteins) of tissue lysates triturated in sterile PBS were applied to confluent Vero cells. Plaques were visualized using Neutral red (Sigma-Aldrich) after 1–3 days of incubation at 37 °C 5% $CO_2$.

**Co-immunoprecipitation**. $1 \times 10^6$ HEK293T cells were transfected with 2 µg of expression plasmids using Lipofectamine 2000. Whole-cell extracts were prepared from transfected cells in lysis buffer (150 mM NaCl, 50 mM Tris pH 7.5, 1 mM EDTA, 0.5% NP40, 10% Glycerol) and were incubated with 50 µl of anti-FLAG magnetic beads for 2 h at 4 °C. Co-immunoprecipitation was performed according to manufacturer's instructions (Anti-Flag Magnetic Beads, Sigma-Aldrich). For co-immunoprecipitation of endogenous proteins $5 \times 10^6$ bone marrow-derived macrophages were infected with CHIKV at a MOI of 10 for 0, 12 and 24 h. The cells were lysed in 1 ml of lysis buffer (150 mM NaCl, 50 mM Tris pH 7.5, 1 mM EDTA, 0.5% NP40, 10% Glycerol, protease inhibitor cocktail). The resultant lysates were cleared by centrifugation at 6,000 g for 10 min at 4 °C. 2 µg rabbit anti-Msr1 IgG was cross-linked to 50 µl of protein A/G agarose beads (ThermoFisher Cat# 20421) with dimethyl pimelimidate (ThermoFisher Cat# 21666). The cleared lysates were incubated with the agarose beads with gentle agitation at 4 °C overnight. The beads were washed five times in ice-cold wash buffer (150 mM NaCl, 50 mM Tris pH 7.5, 1 mM EDTA, 0.5% NP40), and bound proteins were eluted by boiling for 3 min in SDS sample lysis buffer. The bound proteins were resolved by SDS-PAGE, detected by a mouse anti-Msr1/Atg5/Atg12 primary antibody and a secondary HRP-conjugated antibody that only recognizes primary antibodies in their native state (Abcam, Cat# ab131368).

**Immunoblotting and immunofluorescence microscopy**. Immunoblot analysis was done using standard procedures. For immunofluorescence microscopy, after treatment with viruses, cells were fixed with 4% PFA for 30 min. The cells were sequentially permeabilized with 0.5% Triton X-100 for 15 min, blocked with 2% goat serum at room temperature for 1 h, incubated with a primary antibody (10–15 µg/ml) at 4 °C overnight, washed briefly and then incubated with an Alexa Fluor 488/594-conjugated goat anti-rabbit/mouse IgG (1:400, ThermoFisher) for 1 h at room temperature. Nuclei were stained with DAPI. Images were acquired using a Zeiss 880 confocal microscope (objective ×40).

**Statistics and reproducibility**. The sample size chosen for our animal experiments in this study was estimated based on our prior experience of performing similar sets of experiments and power analysis calculations (http://isogenic.info/html/power_analysis.html). All animal results were included and no method of randomization was applied. For all the bar graphs, data were expressed as mean ± s.e.m. A Prism GraphPad Software was used for survival curves, charts and statistical analyses. Survival curves were analyzed using a Log-rank (Mantel-Cox) test. For statistical analyses of in vitro results, a standard two-tailed unpaired Student's t-test was applied. For animal studies, the data were presented as scatter plots with median and an unpaired nonparametric/parametric Mann–Whitney U test was applied to statistical analyses. The results with a p value ≤ 0.05 were considered significant. The sample sizes (biological replicates), specific statistical tests used, and the main effects of our statistical analyses for each experiment are detailed in each figure legend.

**Reporting summary**. Further information on research design is available in the Nature Research Reporting Summary linked to this article.

## Data availability

The datasets generated during and/or analyzed during the current study are available from the corresponding author upon request. Source data underlying figures and tables are provided in Supplementary Data 1. Full blots are shown in Supplementary Information.

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

## Acknowledgements

We are grateful to the Connecticut Agricultural Experiment Station for providing Chikungunya virus. This work was supported in part by a National Institutes of Health grant R01AI132526 to P.W. R.A.F. and E.F. are investigators of the Howard Hughes Medical Institute.

## Author contributions

L.Y., T.G., and G.Y. performed the majority of the experimental procedures. J.M., L.W., H.K., D.Y., T.L., and J.H. contributed to some of the experiments and/or provided technical support. Y.W., S.Z., J.D., F.Y., G.C., A.T.V., R.A.F., and E.F. contributed to data analysis and/or provided technical support. P.W. oversaw the study. L.Y. and P.W. wrote the paper and all the authors reviewed and/or modified the manuscript.

## Competing interests

The authors declare no competing interests.
