## [Peer Review File · Communications Biology]

Reviewers' comments:

Reviewer #1 (Remarks to the Author):

This interesting proposed work of Yang, Geng, Yang et al. addresses the role of MSR1 in antiviral autophagy during infection with CHIKV. The authors report a substantial impact of MSR1 on ATG5-ATG12-ATG16-mediated autophagy and show that this interaction is dependent on the C-terminal FIP200 and WIPI2 binding domain of ATG16L1.

The authors show that MSR1 expression is induced during early infection and that a depletion and deletion of MSR1 leads to an increase of viral load in infected cells and tissues in vivo and in vitro. This is independent of IFN-I response.

They further show that induction of MSR1 leads to an upregulation of autophagy, both indicated by an upregulation of SQSTM1/p62 expression and an increase in LC3 lipidation. They demonstrate that MSR1 directly interacts with ATG5-ATG12 and that this interaction is enhanced by the viral protein nsP1.

Recent literature demonstrates an important role of autophagy in antiviral defence and many viruses have been shown to undermine and manipulate autophagy, often using several simultaneous strategies. Altogether, this work provides a novel mechanism of evasion of the cellular defence by CHIKV, which will help understand both the function of viral factors, as well as the cellular machinery in this process.

While I find this work and the experiments that were performed to support the authors' conclusions to be comprehensive, I have a few minor issues, mainly with some of the western blots.

1. Fig 5A: Some of the lanes (lanes 1-5) don't align with the actin loading control. It looks like at least one lane on the left has been cut off (possibly the DNA ladder?), which then lead to the first band almost missing in the WCE panel.

2. Fig. 5B The actin loading control contains 10 bands, for only 8 samples.

3. In some figures, the notations of mutants are not consistent, e.g. Fig. 4 ATG16L1 vs. ATG16L1-/- or Fig. 6 p62 vs. p62-/-.

4. Fig 6B: It is hard to say whether the reduced band intensity of Cap in the presence of rapamycin (lane 2) is really due to reduced Cap levels in the sample. Because the band intensity on the left seems just as intense as the other Cap bands in the same blot, it is possible that this is a blotting artefact. While the result is probably still credible, it might be best to reload these samples to get a better figure.

5. Similarly, in supplemental fig. S2A, it is likely that the amounts of p62 indeed are not reduced, in *msr1*^{-/-} samples, when compared to the loading control, but repeating this WB with equal protein amounts in all samples would provide a better quality and would then be more convincing.

6. lastly, the manuscript contains a few typing errors, which are minor, but I would like to point out a double-mentioned reference #45 in lane 151, and a missing mention of WB lane 8 in lane 276.

Also, Fig. 6C is not referred to in the text.

Reviewer #2 (Remarks to the Author):

This study aims to elucidate the physiological function of Msr1 during CHIKV infection. The data suggest that Msr1 inhibits CHIKV replication by promoting autophagy. This work is fascinating, and

the findings support the idea that Msr1 participates in antiviral immunity.

1. Fig.1, the aggravated injury was found in Msr1^{-/-} mice compared to WT mice. Was there any difference in autophagy activation in the macrophages from damaged tissue?
2. Fig.2, the fluorescent probes, DALGreen or DAPGreen, would be better for monitoring autophagy.
3. The author claimed that CHIKV nsP1 interacts with Msr1 and enhances Msr1-Atg12 interaction. Does nsP1 also interact with Atg12, and Msr1 influence their interaction? Also, does Msr1 influence the nsP1 localization in autophagosome?
4. Which domain of Msr1 involved in the interaction with Atg12? Does Msr1 interact with CHIKV nsP1 and Atg12 through the differential domain?

RE: COMMSBIO-20-0583A

Faten Taki, PhD.
Associate Editor

Dear Dr. Taki,

We thank you for giving us a chance to revise the manuscript COMMSBIO-20-0583A. We thank both reviewers for their constructive opinions, recognition of the importance and novelty of the manuscript. We have now comprehensively addressed the reviewers' critiques.

Major changes include: 1) moving original Fig.2A to Fig.1H; 2) addition of new Fig.2G, H, Fig.5E, Fig.6B, C, D; 3) replacement of Fig.6E, S2A by improved ones; 4) conversion of bar graphs into box-and-whisker or dot-plot format; 5) inclusion of uncropped immunoblots and source data.

Reviewer #1 (Remarks to the Author):

This interesting proposed work of Yang, Geng, Yang et al. addresses the role of MSR1 in antiviral autophagy during infection with CHIKV. The authors report a substantial impact of MSR1 on ATG5-ATG12-ATG16-mediated autophagy and show that this interaction is dependent on the C-terminal FIP200 and WIPI2 binding domain of ATG16L1. The authors show that MSR1 expression is induced during early infection and that a depletion and deletion of MSR1 leads to an increase of viral load in infected cells and tissues in vivo and in vitro. This is independent of IFN-I response. They further show that induction of MSR1 leads to an upregulation of autophagy, both indicated by an upregulation of SQSTM1/p62 expression and an increase in LC3 lipidation. They demonstrate that MSR1 directly interacts with ATG5-ATG12 and that this interaction is enhanced by the viral protein nsP1. Recent literature demonstrates an important role of autophagy in antiviral defence and many viruses have been shown to undermine and manipulate autophagy, often using several simultaneous strategies. Altogether, this work provides a novel mechanism of evasion of the cellular defence by CHIKV, which will help understand both the function of viral factors, as well as the cellular machinery in this process. While I find this work and the experiments that were performed to support the authors' conclusions to be comprehensive, I have a few minor issues, mainly with some of the western blots.

1. Fig 5A: Some of the lanes (lanes 1-5) don't align with the actin loading control. It looks like at least one lane on the left has been cut off (possibly the DNA ladder?), which then lead to the first band almost missing in the WCE panel.

Response: Yes. The protein ladder was cut off. The actin has been re-aligned.

2. Fig. 5B The actin loading control contains 10 bands, for only 8 samples.

Response: This has been corrected.

3. In some figures, the notations of mutants are not consistent, e.g. Fig. 4 ATG16L1 vs. ATG16L1^{-/-} or Fig. 6 p62 vs. p62^{-/-}.

Response: These have been corrected.

4. Fig 6B: It is hard to say whether the reduced band intensity of Cap in the presence of rapamycin (lane 2) is really due to reduced Cap levels in the sample. Because the band intensity on the left seems just as intense as the other Cap bands in the same blot, it is possible that this is a blotting artefact. While the result is probably still credible, it might be best to reload these samples to get a better figure.

Response: CHIKV Cap has been shown to be degraded by autophagy⁴⁷. We recognize that this discrepancy is likely due to poor immunoblotting techniques or rapamycin quality. We repeated this experiment and presented a new figure (**Fig.6E**).

5. Similarly, in supplemental fig. S2A, it is likely that the amounts of p62 indeed are not reduced, in *msr1*^{-/-} samples, when compared to the loading control, but repeating this WB with equal protein amounts in all samples would provide a better quality and would then be more convincing.

Response: We agree. P62 was not significantly reduced in *Msr1*^{-/-} macrophages after CHIKV infection. We repeated WB with similar loading controls (**Fig.S2A**).

6. lastly, the manuscript contains a few typing errors, which are minor, but I would like to point out a double-mentioned reference #45 in lane 151, and a missing mention of WB lane 8 in lane 276. Also, Fig. 6C is not referred to in the text.

Response: We have carefully proof-read and these errors have been corrected (p6, line 137; p9, line 229; Fig6C, now Fig.6E mentioned on p11, line 259).

Reviewer #2 (Remarks to the Author):

This study aims to elucidate the physiological function of Msr1 during CHIKV infection. The data suggest that Msr1 inhibits CHIKV replication by promoting autophagy. This work is fascinating, and the findings support the idea that Msr1 participates in antiviral immunity.

1. Fig.1, the aggravated injury was found in *Msr1*^{-/-} mice compared to WT mice. Was there any difference in autophagy activation in the macrophages from damaged tissue?

Response: Purification of macrophages from infected mouse feet involves many steps (either positive or negative selection, washing etc.) that may activate autophagy non-specifically. We thus analyzed LC3-II conversion in whole mouse feet by immunoblotting. **Fig.2H** shows that LC3-I to LC3-II conversion was induced in WT mouse tissues after CHIKV infection and the LC3-II levels were lower in *Msr1*^{-/-} than WT mice.

2. Fig.2, the fluorescent probes, DALGreen or DAPGreen, would be better for monitoring autophagy.

Response: We observed consistent results using DAPGreen in WT and *MSR1*^{-/-} cells infected with CHIKV (**Fig.2G**).

3. The author claimed that CHIKV nsP1 interacts with *Msr1* and enhances *Msr1*-Atg12 interaction. Does nsP1 also interact with Atg12, and *Msr1* influence their interaction? Also, does *Msr1* influence the nsP1 localization in autophagosome?

Response: Indeed ATG12 can co-immunoprecipitate with recombinant nsP1 (**Fig.6B**) or endogenous nsP1 during CHIKV infection (**Fig.6C**), and MSR1 indeed enhances nsP1-ATG12 interaction (**Fig.6B**). Both LC3 punctae and nsP1 localizations to LC3-positive autophagosomes were reduced in *MSR1*^{-/-} cells during CHIKV infection (**Fig.6D**). These data suggest that nsP1-MSR1-ATG12 forms a complex.

4. Which domain of Msr1 involved in the interaction with Atg12? Does Msr1 interact with CHIKV nsP1 and Atg12 through the differential domain?

Response: The domain of MSR1 that mediates MSR1-ATG12 interaction is its intracellular 1-50aa (**Fig.3E**). While 1-50aa and 51-end of MSR1 binds endogenous nsP1 very weakly, 1-80aa with the transmembrane domain binds nsP1 much better (**Fig.5E**), suggesting that MSR1 interaction with nsP1 requires both 1-50aa and the membrane anchorage of MSR1. Future efforts are needed to further pinpoint the exact sequence within 1-80aa of MSR1 that interacts with nsP1 and ATG12 respectively.

We have modified the text to reflect the current state of results and hope that the above clarification will satisfy both reviewers. We thank you again for the efforts on the manuscript and look forward to your favorable consideration in light of the new evidence.

Sincerely

Penghua Wang

REVIEWERS' COMMENTS:

Reviewer #1 (Remarks to the Author):

The Authors have repeated and corrected the requested experiments and text passages. I now recommend this article for publication.

Reviewer #2 (Remarks to the Author):

I have no further comments and recommend acceptance.